# Single-Center Retrospective Analysis of Paraneoplastic Syndromes with Peripheral Nerve Damage

**DOI:** 10.3390/brainsci12121656

**Published:** 2022-12-02

**Authors:** Jing Tian, Cuifang Cao, Ruihan Miao, Haoran Wu, Kun Zhang, Binbin Wang, Zhou Zhou, Ruomeng Chen, Xiaoyun Liu

**Affiliations:** 1Department of Neurology, The Second Hospital of Hebei Medical University, Shijiazhuang 050061, China; 2Neuroscience Research Center, Medicine and Health Institute, Hebei Medical University, Shijiazhuang 050011, China

**Keywords:** paraneoplastic neurological syndrome, peripheral nerve, electrophysiological, age

## Abstract

There are few clinical and electrophysiological studies on paraneoplastic neurological syndrome (PNS) with peripheral nerve damage, which brings great challenges to clinical identification and diagnosis. We analyzed the clinical and electrophysiological data of twenty-five confirmed PNS cases using peripheral nerve damage patients. The results showed the most common chief complaint was weakness (20/25, 80%), followed by numbness (13/25, 52%). Nineteen patients (76%) exhibited peripheral nervous system lesions prior to occult tumors, and the median time from symptom onset to the diagnosis of a tumor was 4 months. The electrophysiological results revealed a higher rate of abnormal amplitudes than latency or conduction velocity, especially in sensory nerves. Meanwhile, we found that, compared with patients >65 y, patients aged ≤65 y exhibited more chronic onset (*p* = 0.01) and longer disease duration (*p* = 0.01), more motor nerve involvements (*p* = 0.02), more amplitude involvement (*p* = 0.01), and higher rates of the inability to walk independently at presentation (*p* = 0.02). The present study construed that weakness and paresthesia are common symptoms in PNS with peripheral nerve damage in some areas, and the electrophysiological results mainly changed in amplitude. Tumor screening in young and middle-aged patients with peripheral neuropathy cannot be ignored.

## 1. Introduction

Paraneoplastic neurological syndrome (PNS) is a rare autoimmune disorder that occurs in approximately 0.1% of cancer patients, and the incidence has progressively increased [1,2,3]. PNS can involve the central or peripheral nervous systems, autonomic systems, neuromuscular junctions, or muscles. PNS includes disorders that occur prior to or during cancer due to the remote effect of the tumor independent of neoplastic infiltration, cancer treatment, infectious and metabolic complications, or other well-known causes of neuropathy [4]. Neuropathic symptoms precede the detection of cancer in a majority of cases; therefore, the definitive diagnosis of paraneoplastic neuropathy is crucial to enabling the early detection of malignancies and the immediate commencement of therapy for cancer, as well as for the paraneoplastic neuropathy [5].

PNSs with peripheral nerve damage are extremely rare compared with the well-characterized limbic encephalitis and paraneoplastic cerebellar degeneration; further, they are difficult to differentiate from typical peripheral neuropathy. Most of the current case analyses have focused on the clinical and electrophysiological features of typical syndromes with specific antibodies [6,7,8], tumor types [9], or rare case reports [10,11]. However, paraneoplastic neuropathy clinically occurs with extensive variation in the progression of the neuropathy; its pattern; the degree of sensory, motor, and autonomic involvement; and the presence or absence of paraneoplastic autoantibodies [12]. These factors severely hinder the diagnosis and prompt treatment of the disease. Therefore, this study summarized all confirmed paraneoplastic syndromes requiring electrophysiological examination in the Second Hospital of Hebei Medical University, China, in recent years. Analyzing these patients’ clinical and electrophysiological characteristics contributes to increasing the awareness of these disease spectrums, their prompt diagnosis by clinical physicians, and improved patient outcomes.

## 2. Materials and Methods

### 2.1. Participants

Between January 2014 and June 2020, 94 probable or definite PNS cases were retrospectively assessed at the Neurology Department of the Second Hospital of Hebei Medical University in Northern China. Patients were diagnosed by experienced physicians according to the widely accepted criteria described by Graus et al. [13]. Among them, 57 patients who reported symptoms in the intracranial central nervous system and did not require neuroelectrophysiological examination were excluded. We selected only 37 patients who needed neurophysiologic examination, 25 of whom had definite PNS, while the rest had possible PNS. Clinical data of the 25 patients, including age, sex, course of disease, chief complaint, accompanying symptoms and signs, smoking and drinking status (current and sustained consumption for ≥30 years), sequence and interval of tumor and syndrome, time from symptom onset to first visit and diagnosis, typical or atypical paraneoplastic syndrome, modified Rankin Scale, cancer type, onconeural antibody, tumor markers, and autoantibodies were analyzed retrospectively. The study protocol was approved by the human ethics committee of the Second Hospital of Hebei Medical University, PR China.

### 2.2. Electrophysiology

Twenty of the twenty-five patients underwent traceable neuroelectrophysiological examinations traceable to our hospital. Motor and sensory conduction studies were performed to measure the amplitude and duration of the negative peak of the compound muscle action potential (CMAP) with distal (dCMAP) and proximal (pCMAP) stimulation; conduction velocity (CV) of the motor (MCV); and distal motor latency (DML) of the median, ulnar, peroneal, and tibial nerves. Furthermore, the sensory nerve action potential (SNAP) and sensory conduction velocity (SCV) of the median, ulnar, and peroneal nerves were recorded.

### 2.3. Laboratory Examinations

Serum tumor markers, including alpha-fetoprotein (AFP); AFP-heterogeneity L3 (AFP-L3); AFP-L3/AFP; carcinoembryonic antigen (CEA); carbohydrate antigens 125, 153, 724, and 199 (CA125, CA153, CA724, CA199); cytokeratin 19 fragments (CY21-1); neuron-specific enolase; total prostate-specific antigen (tPSA); and free prostate-specific antigen (fPSA) were detected in 19 of 25 patients using electrochemiluminescence immunoassays. Well-characterized onconeural antibodies in serum and cerebrospinal fluid were evaluated by immunoblotting, immunofluorescence, or immunohistochemistry in eight patients, including anti-Hu-IgG, anti-Yo-IgG, anti-Ri-IgG, anti-CV2-IgG, anti-Ma-IgG, anti-amphiphysin-IgG, anti-ANNA-3-IgG, anti-Tr-IgG, anti-PCA-2-IgG, and anti-GAD-IgG.

### 2.4. Statistical Analysis

Statistical analyses were performed using SPSS version 25.0. Categorical data were presented as proportions, while continuous data were presented as means and standard deviations or means and interquartile ranges depending on their distribution. Differences in proportions were tested using the x2 test. The numerical variables were analyzed using the Kruskal–Wallis H test, and *p* < 0.05 was considered to be significant.

## 3. Results

We retrospectively evaluated 25 definitively diagnosed PNS cases from 37 patients. The demographic data and clinical details of the 25 patients were obtained from their medical records (Table 1). The mean age at onset was 61.68 years (standard deviation 9.23; range, 36–75), and the ratio of males to females was 11.5 (23 men and 2 women). Two patients (8%) deteriorated rapidly within two weeks and presented with acute onset, whereas the other twenty-three patients (92%) presented with subacute (2 weeks–3 months; 9, 36%) or chronic (>3 months, 14, 56%) disease onsets. Most patients (19/25) lacked a known neoplasm at the onset of the syndrome, and a new neoplasm diagnosis was established within a median of 4 months (range, 0.5–336). Six patients with known neoplasms developed PNS while being treated for the newly discovered metastatic disease in a median of 2.5 months (range, 0.5–312). The median time between the symptom onset and seeking medical advice was 2 months (range, 0–324), but from onset to diagnosis it was 3 months (range, 0.5–336). In the appendix, we describe the clinical features, onconeural antibodies, and tumor markers of these patients.

Twenty of the twenty-five patients underwent neuroelectrophysiological examination. In total, 182 nerves were tested, including 102 motor nerves and 80 sensory nerves. Meanwhile, we compared the abnormality of each parameter and found no significant differences in the abnormality rate of the motor and sensory nerves, regardless of whether the upper or lower limbs were distinguished. However, the abnormal rate of amplitude was significantly higher than the abnormal rate of the latency period or CV, whether in the motor or sensory nerves. In addition, patients with subacute sensory neuronopathy exhibited more sensory nerve involvement than patients with other syndromes, and their SNAP and SCV scores were lower (see Appendix A).

To further clarify the relationship between age and disease course, we divided the patients into two groups: Group1—A ≤65 y; Group2—A >65 y. The two groups were not significantly different in terms of sex and medical history, such as diabetes. We then analyzed the relationship between age and the course of the disease, the severity of the disease, the time from symptom onset to seeking medical advice, and peripheral nerve involvement (Table 2). The results indicated that age was significantly related to the course of the disease, the time from symptom onset to seeking medical advice, and the mRS score. In other words, the longer time from onset to consultation for patients ≤65 y, the more insidious the onset, the more insufficient attention to the disease, and the worse the ability to live independently compared with patients >65 y. The results revealed that age was significantly related to motor nerve damage rate, especially the dCMAP. This might have been related to the long period between onset and admission.

## 4. Discussion

In this study, we retrospectively analyzed the clinical and neuroelectrophysiological characteristics of 25 PNS patients with peripheral nerve damage. The most common paraneoplastic syndromes were subacute sensory neuronopathy (12/25, 48%) and chronic sensorimotor neuropathy (5/25, 20%). Lung carcinoma was the most common primary tumor (15/24, 62.5%). Most patients were unable to walk independently when seeking medical advice and had mRS scores >2 (21/25, 84%). The electrophysiological results showed that the rate of abnormal amplitudes was significantly higher than the latency or CV in the sensory and motor nerves. Meanwhile, compared with patients aged >65 y, those aged ≤65 y exhibited a longer time from onset to consultation, more insidious onset, less sufficient attention to disease, and worse ability to live independently.

The spectrum of paraneoplastic autoimmunity has dramatically expanded following the update of detection methods and the deepening of understanding [14,15]. As the disease spectrum changes, so do the patient’s complaints. In our cohort, the most common complaint was weakness, accounting for 80%, which is different from the conclusion of the main complaint of paresthesia in studies from other countries [16,17,18]. However, in our patients, the positivity rate (50%) and type of onconeural antibodies were not significantly different from those in previous studies, and the anti-Hu antibody was still the most common type of antibody [1,19].

Subacute sensory neuronopathy was still the most common type of neuronopathy. These results are consistent with the findings of a cohort study conducted in our country, excluding central nervous system damage [19]. Another common type of neuropathy in our patients was chronic sensorimotor neuropathy. According to the consensus paper on “classical” and “non-classical” syndromes in 2004 [13], this subtype has been classified as a “non-classical” syndrome. However, it has been pointed out in prior studies that peripheral nerve damage is the most commonly diagnosed “non-classical” syndrome, and subacute or chronic sensorimotor neuronopathy is the most frequent “non-classical” syndrome [20]. Clinically, when paraneoplastic syndrome cannot be diagnosed because of the diagnosis of subacute or chronic sensorimotor neuropathy, regular follow-up observation should be preferred. Most patients had subacute or chronic onset (23/25, 92%), and the symptoms were mild and insidious at the beginning and were easily ignored. However, as the disease progresses, the symptoms gradually worsen, which is consistent with the manifestations of peripheral neuropathy due to other causes, and distinguishing becomes difficult [21]. In our cohort, most patients were unable to walk independently at presentation (21/25, 84%), and the time from symptom onset to diagnosis was as long as 3 months, which seriously delayed treatment. Although we have not tracked the long-term prognosis of patients, a large number of studies have shown that whether for paraneoplastic syndrome or the tumor itself, early identification, diagnosis, and effective treatment are the best options to improve the prognosis of patients [22].

We analyzed the electrophysiological outcomes of the patients (20/25). As no classification criteria exist for axonal and demyelination for PNS, the patients were not classified, but we compared the differences in sensory and motor nerves, upper and lower limb nerves, subacute sensory neuropathy, and other neuropathies and the corresponding parameters of the individual nerves. Our results suggest that amplitude abnormalities are more common, especially in the nerves of the lower extremities and sensory nerves, which also corresponded to the fact that most patients were unable to walk independently. In motor nerves, both amplitude and CV are susceptible to involvement. Few studies have been conducted on the electrophysiological aspects of paraneoplastic peripheral neuropathy. Some studies of anti-Hu-antibody-related paraneoplastic peripheral neuropathy have shown a similar electrophysiological performance [7]. The motor nerve abnormalities suggest that the pathological process of subacute sensory neuropathy associated with anti-Hu syndrome is frequently not restricted to the dorsal root ganglia and is probably more complex [23]. The mechanisms of the nerves of the lower extremities and motor fiber involvement in our patients remain unclear. As Hu proteins are not present in peripheral nerves, other nerve proteins may be the targets of the paraneoplastic immune process [24].

As per our results, patients aged ≤65 years had a longer disease course, later seeking of medical advice, lower amplitude, and more severe disease than the patients aged >65 years. In addition, the ≤65 y patients had a higher rate of motor nerve involvement, which further aggravated their condition. This might have been due to the higher degree of tolerance of the disease in ≤65 y patients, especially for peripheral neuropathy with a chronic or subacute onset, which is easily overlooked and not considered abnormal in the body, or it is misdiagnosed as ordinary peripheral neuropathy. Previous studies have shown that early diagnosis and treatment can improve patient outcomes; however, no clinical or electrophysiological differences in age have been reported [1,25]. Our results indicate that awareness of this disease should be raised among people aged ≤65 y in the future.

Our study had several limitations. First, it was a retrospective, observational, single-center study; therefore, it had a certain level of inherent bias and decreased the generalizability of the study results. Second, our patients were not routinely tested for paraneoplastic antibodies; therefore, some patients were excluded because they could not be diagnosed with a confirmed PNS, resulting in a small sample size. Third, we did not have a long-term follow-up of the patients’ prognoses. Finally, the PNS-Care panel updated the diagnostic criteria for PNS on the basis of new phenotypes and antibodies [4], but the research revealed that the new PNS-Care Score provides a specific and strict approach to improving diagnostic accuracy. However, its increased risk of underdiagnosis is a cause for concern [26]. Some of the patients in our cohort were not tested for antibodies, as we used the 2004 diagnostic criteria [13].

## 5. Conclusions

Subacute sensory neuropathy is the most common paraneoplastic nerve syndrome with peripheral nerve damage, and weakness and paresthesia are common symptoms. The electrophysiological results mainly changed in amplitude, and most patients were unable to walk independently at presentation. Patients aged ≤65 years have a longer interval from onset to medical advice, which seriously delays diagnosis and treatment. The awareness of peripheral neuropathy needs to be improved in the future.

## Figures and Tables

**Table 1 brainsci-12-01656-t001:** Baseline characteristics of PNS with peripheral nerve damage.

Demographics	Value	Demographics	Value
Age (Mean ± Standard Deviation)	61.68 ± 9.23	Time from symptom onset to diagnosis (months)	3 (10.5)
Sex (male: female)	23:2	Time from onset to first visit (months)	2 (5)
Course of disease		Typical PNS	14 (25) 56%
Acute	2 (8%)	LEMS	1 (14) 7.1%
Subacute	9 (36%)	Dermatomyositis	1 (14) 7.1%
Chronic	14 (56%)	SSN	12 (14) 85.7%
Chief complaint with myasthenia (%)	20 (80%)	With myasthenia	10 (12) 83.3%
Chief complaint with paresthesia (%)	13 (52%)	Atypical PNS	11 (25) 44%
Chief complaint with myasthenia and paresthesia (%)	8 (32%)	CSMN	5 (11) 45.5%
Accompanying symptoms and signs		Motor neuron disease	3 (11) 27.3%
Dysarthria	5 (20%)	GBS	2 (11) 18.2%
Dysphagia	4 (16%)	Myasthenia gravis	1 (11) 9.1%
Decreased tendon reflexes	15 (60%)	Onconeural antibody	8 (32%)
Myalgia	9 (36%)	Hu (+)	3 (8) 37.5%
Smoker (%)	10 (40%)	Amphiphysin (+)	1 (8) 12.5%
Alcoholics (%)	5 (20%)	Abnormal specific tumor markers	9 (19) 47.4%
The neoplasm occurred before the syndrome (%)	6 (24%)	Autoantibodies	18 (25) 72%
Interphase, months, median (range)	2.5 (0.5–12)	ANA	10 (18)
The neoplasm occurred after the syndrome (%)	19 (76%)	SSA-52 KDa	1 (18)
Interphase, months, median (range)	4 (0.5, 36)	SSA-60 KDa	1 (18)

LEMS, Lambert–Eaton myasthenic syndrome; GBS, Guillain–Barré syndrome; ANA, antinuclear antibody; SSN, subacute sensory neuronopathy; PNS, paraneoplastic neurological syndrome; CSMN, chronic sensorimotor neuropathy.

**Table 2 brainsci-12-01656-t002:** Differences in disease course and electrophysiological results by age.

Group	1	2	X2/U	*p*
Age	≤65 y	>65 y		
*n*	15	10		
Slow onset (>3 m)	12/15, 80.0%	2/10, 20%	8.42	0.01
CD (m)	11 (9)	2 (1.8)	30.50	0.01
TSOSMA (m)	3 (5)	0.5 (2)	39.00	0.08
Typical syndrome or not	6/15 (40%)	8/10 (80%)	2.44	0.10
mRS (≥3)	15/15, 100%	4/10, 55.6%	6.86	0.02
Lung cancer	8/15 (53.3%)	8/10 (80%)	1.78	0.23
NSOBT	10/15 (66.7%)	9/10 (90%)	0.74	0.35
Peripheral nerve research				
*n*	11	9		
Sensory nerve involvement rate	1.00 (1.00)	0.25 (1.00)	43.50	0.66
SNAP	1.00 (1.00)	0.25 (0.75)	39.00	0.46
SCV	0.99 (1.00)	0.00 (0.50)	39.00	0.55
Motor nerve involvement rate	1.00 (0.50)	0.25 (0.67)	18.00	0.02
DML	0.00 (0.92)	0.00 (0.00)	33.00	0.23
dCMAP	1.00 (0.71)	0.00 (0.50)	17.50	0.01
MCV	0.33 (0.79)	0.17 (0.67)	28.00	0.11
LENIR	0.40 (0.30)	0.20 (0.40)	25.50	0.07
LENIMNI	1.00 (0.50)	0.25 (1.00)	26.00	0.08
LENISNI	1.00 (1.00)	0.50 (1.00)	44.00	0.71
dCMAP of median nerve	2.20 (3.20)	5.10 (1.65)	14.50	0.01
dCMAP ulnar nerve	1.90 (4.45)	7.00 (3.70)	18.50	0.02
dCMAP of peroneal nerve	0.50 (3.40)	2.80 (2.08)	26.00	0.08

CD, course of disease; TSOSMA, time from symptom onset to seeking medical advice; mRS: modified Rankin Scale; NSOBT, neurological syndrome occurs before tumor; SNAP, sensory nerve action potential; SCV, sensory conduction velocity; DML, distal motor latency; dCMAP, distal compound muscle action potential; MCV, motor conduction velocity; LENIR, lower extremity nerve involvement rate; LENIMNI, lower extremity nerve involvement in the motor nerves involved; LENISNI, lower extremity nerve involvement in the sensory nerves involved.

## Data Availability

The data presented in this study are available upon request from the corresponding author.

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
