# Peer review of "Single-Center Retrospective Analysis of Paraneoplastic Syndromes with Peripheral Nerve Damage"

_brainsci, 2022, doi:10.3390/brainsci12121656_

Round 1

Reviewer 1 Report

An interesting study about peripheral neuropathies as a paraneoplastic syndrome. The topic is intriguing, whereas the main limitation (as the authors clearly state) is represented by the very small sample size (25 patients); moreover, also the subdivision into two groups  according to the age (above/below 65 years) appears really arbitrary

 Why only 25 among 25 patients were submitted to neurophysiological examination?

The authors should avoid redundant sentences, as "The spectrum of paraneoplastic autoimmunity has dramatically expanded following the update of detection methods and the deepening of understanding. The spectrum of paraneoplastic neuropathies has increased to encompass motor neuropathies, small fiber neuropathies, and autonomic and nerve hyperexcitability syndromes" (lines 154-156)

Do any patients undergo an ultrasonographic examination to evaluate indirect signs related to the EMG/ENG findings?

Reviewer 2 Report

Here are my thoughts, wrapped in the comment boxes next to the text/manuscript.  Revision is required before I can consider reviewing it again, including having it written up by native language speakers, if that can be helped. The topic is certainly interesting, but it has to be written in a more compelling light.

  Main issues are   
  1. How did they come up with the 'Ab panel' that they used ? Has it been validated before in ANY study ?
  2. Confounding factors include that they included myasthenia gravis and Lambert Eaton syndrome.
  3. 60% of patients had alcohol abuse and were smokers. How were these patients accounted for ?

Reviewer 3 Report

The ms by Tian et al. presents the analysis of patients with peripheral nerve damage in the course of paraneoplastic syndrome. The subject is gaining more and more attention, with longer oncological patients' survival and a tendency to appear more and more frequently. Therefore widening comprehensive knowledge of different aspects of paraneoplastic syndromes is necessary. The analysis is well conducted from the clinical point of view, and the results are clearly and adequately presented.

However, the research presented has some weaknesses, like retrospectives, the number of patients included, and lack of a control group.

Moreover, even if the English used is understandable, there are a number of grammatical, spelling, and punctuation mistakes - needs one thorough reading.

Round 2

Reviewer 1 Report

Dear authors, my concerns about the sample size and the study design are still ongoing. Therefore, I think that the manuscript in the current form is not suitable for publication.

Author Response

Thank you very much for your comments. 

Due to the rarity and wide spectrum of PNS, our study had the limitations of small sample size, retrospective review and no control group. However, there have been few clinical and electrophysiological studies on paraneoplastic peripheral neuropathy in the past. Our findings suggest that the chief complaint of weakness and young and middle-aged patients should also pay attention to tumor screening.

We look forward to your approval and support.

Reviewer 3 Report

The authors implemented the necessary changes and included sufficient explanations in the corrected manuscript and response chart.

Author Response

Thank you very much for your comments.

We look forward to your approval and support.